# PACR: Pixel Attention in Classification and Regression for Visual Object Tracking

**Da Li, Haoxiang Chai, Qin Wei \*, Yao Zhang and Yunhan Xiao**

School of Information Engineering, Wuhan University of Technology, Wuhan 430070, China;
frankli26@whut.edu.cn (D.L.); hx453794261@whut.edu.cn (H.C.); zyao@whut.edu.cn (Y.Z.);
xzxz21@whut.edu.cn (Y.X.)
**\*** Correspondence: qinwei@whut.edu.cn

**Abstract:** Anchor-free-based trackers have achieved remarkable performance in single visual object tracking in recent years. Most anchor-free trackers consider the rectangular fields close to the target center as the positive sample used in the training phase, while they always use the maximum of the corresponding map to determine the location of the target in the tracking phase. Thus, this will make the tracker inconsistent between the training and tracking phase. To solve this problem, we propose a pixel-attention module (PAM), which ensures the consistency of the training and tracking phase through a self-attention module. Moreover, we put forward a new refined branch named Acc branch to inherit the benefit of the PAM. The score of Acc branch can tune the classification and the regression of the tracking target more precisely. We conduct extensive experiments on challenging benchmarks such as VOT2020, UAV123, DTB70, OTB100, and a large-scale benchmark LaSOT. Compared with other anchor-free trackers, our tracker gains excellent performance in small-scale datasets. In UAV benchmarks such as UAV123 and DTB70, the precision of our tracker increases 4.3% and 1.8%, respectively, compared with the SOTA in anchor-free trackers.

**Keywords:** object tracking; anchor free; pixel attention; refined branch

**MSC:** 68T01; 68T07; 68T45





## 1. Introduction

Single Object Tracking is a fundamental work in computer vision with so many applications in various fields such as autonomous driving, human–computer interactions, video annotation [1–3], etc. Although brilliant methods have been proposed during the past decade, there are still some difficulties in practical applications due to challenging factors, such as fast motions, background clutter, occlusions, deformation, and other difficulties, which urges us to develop trackers with strong adaptiveness and robustness.

In recent years, no matter if they are transformer-based or Siamese-based, or correlation-based, most trackers which adopt the standard tracking-by-detection framework are divided into two subtasks: feature extraction and object location. The feature extraction subtask obtains the feature which helps to distinguish the differences and similarities between the template frame and the subsequent search frame. While the object location subtask will give the location of the tracking object with the help of the result computed by the previous feature's extraction subtask. Usually, the computing method is correlation in the Discriminative Correlation Filters (DCFs) [4–9] or convolution in the Siamese trackers [10–15]. Recently, transformer-based trackers [16–20] such as OSTrack [21] adopt the dot product as the computing method. Those three computing methods are utilized to calculate the above similarities.

In the tracking phase, the tracker could get the target's approximate location by the maximum value of the calculated similarities. But in the training phase, most trackers only can use the proximity principle, which means they have to regard the rectangle area near

the target as the ground truth shown in Figure 1, especially the anchor-free trackers. It can be found that the location rules are inconsistent between the training and tracking phase.

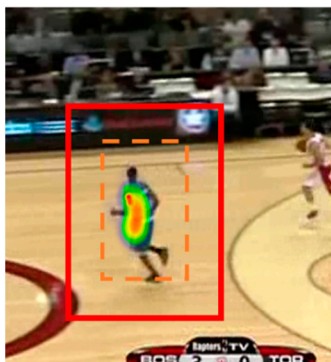 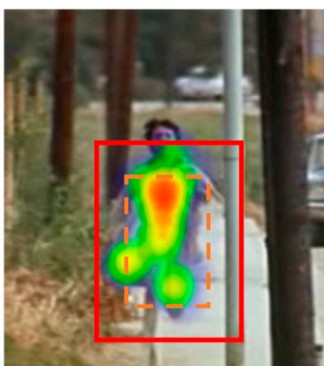

**Figure 1.** The current method of determining the tracking target is shown by the orange dash line (all points close to the tracking center are positive samples), which contains a lot of background information that will cause the tracker to perform wrong learning during the training phase. Pixel-attention encodes each pixel and calculates the weight, suppressing the small correlation score when generating positive sample labels.

The Pixel Attention Classification and Regression (PACR) tracker is proposed to solve the inconsistencies. With the above-mentioned tracking methods, our tracker has remarkable performance in some challenging scenarios. Our contribution can be summarized in three aspects:

1. A new attention is proposed, named pixel attention, to tackle the incongruence between the training and tracking phases, which differs from the channel and spatial attention. The pixel attention makes the tracker focus on the object which is most like the target to adjust the ground truth and reduce the training time.
2. The Acc branch is put forward, which will further expand the advantages of pixel attention during the tracking phase to improve the tracking result in the object location subtask. The score of the Acc which is trained by the predicting acceleration can rectify the tracker's outputs more accurately.
3. A novel end-to-end tracker is proposed with anchor-free baseline. Using the joint classification and regression to compute the loss score in the training phase, the score of the Acc branch can be better integrated into the Cls (classification) branch and the Reg (regression) branch. As a result, the quality of the tracker can be improved notably.

## 2. Related Work

This section will briefly review some related work about single visual object tracking. Mainly, some strategies are introduced, such as anchor-free mechanisms and attention in object tracking.

### 2.1. Discriminant Correlation Filter

DCF-based tracker learning a target model to locate a target by minimizing an objective has attracted much attention in object tracking because of its finite computation cost and robustness. MOSSE [22] (Minimum Output Sum of Squared Error filter) is the first to introduce the correlation algorithm into object tracking. Then CSK [11] and the Kernel Correlation Filter [5] (KCF), which introduced the ridge regression and approximate dense sampling method based on cyclic shifts into the tracker, made DCF-based trackers more popular. SRDCF [6], STRCF [7], and ARCF [8] introduce the spatial penalty and temporal penalty to solve the boundary effect and improve the robustness and accuracy of the tracker Auto-track [9], and this [23] makes the UAV-based object track become the practical

application stage from the research stage; the latter also summarizes the problems existing in the current DCF-based UAV trackers.

### 2.2. Siamese-Based Tracker

Siamese-based trackers have attracted attention recently and have become very popular in this field. The first one proposed to solve the tracking problem is the SiamFC [10], tracking the target by learning the similarity between the template and search region. Inspired by SiamFC, many works have followed and put forward updated models. The correlation filter layer and online tracking are introduced into the Siamese-based tracker to improve the accuracy in CFNet [24]. DaSiam [13] successfully explored the effect of sample selection on the tracker, which improves the robustness of the tracker. With the development of RPN networks, SA-Siam [11] built a two-fold Siamese network with a semantic branch and an appearance branch. They are trained separately to keep the heterogeneity of features but tested in a combination way. SA-Siam has proved that the training phase can differ from the testing phase. SiamPRN++ [14], using ResNet-50 as the backbone, has declared that feature extraction may affect the tracking result. Meanwhile, it randomly shifts the object location in the search region during training to eliminate the center bias, surprisingly improving tracking accuracy.

### 2.3. Anchor-Free Mechanism

Although anchor-based trackers have many benefits, such as boosting the tracking result, improving the tracking accuracy, or making the IOU score higher, some difficulties of anchor-based mechanisms still need to be solved. For example, the performance of most anchor-based trackers is susceptible to the hyper-parameter, and the size and the aspect ratio of the bounding box of these trackers are fixed. Moreover, the design of anchors is also related to the object category. In recent years, the anchor-free mechanism has gained more and more attention because of its simplicity and efficiency. The anchor-free mechanism doesn't have trouble like the anchor-based mechanism in anchor design. It was first put forward in DenseBox to solve the object detection, then CornerNet [25] and ExtremeNet [26] further studied the anchor-free mechanism, expanding the predicted corner to 2 or 4 points. FCOS [27] directly indicates the bounding box with the help of a new strategy named center-ness. Ocean [28], SiamFC++ [15], and SiamCAR [29] are the anchor-free trackers, calculating the loss by IoU-loss in the training phase, which increase the accuracy of the predicted bounding box.

### 2.4. Attention in Object Tracking

Nowadays, attention has gained more and more attraction in single visual object tracking and achieved impressive results. Most attention-based trackers apply attention mechanisms in two ways. Guided by the principles of CNN, the attention mechanism can be a backbone to extract features. For example, OSTrack [21] directly uses ViT [30] as the backbone to extract features; Swin [31], PVT [32], PVTv2 [33], and MixFormer [17] also adopt the same strategy. They prove an attention-based backbone can gain the same benefit as a traditional CNN-based backbone. However, the CNN-based backbone performs better if it takes the similar architecture and strategies of a recent study named ConvNeXt [34]. Merely fusing the features extracted from the backbone is the other method. TCTrack [18] delivers the information from the last frame to the current search frame with the help of attention. STMTrack [16] directly applies a self-attention encoder to the computation of the search-frame feature by combining the historic search frames. AiATrack [20], ToMP [35], and Transformer Tracking [36] are also this way. Although they can track targets robustly without online learning, padding design is difficult to initialize for direct use, such as Stark, which will incur enormous computational costs. Considering this, we propose a new method to combine the attention mechanism and improve the tracker's accuracy.

*2.5. Guidance*

In recent work of anchor-free Siamese trackers, most trackers select the rectangle region close to the center of the target as a positive target during training. It is not a good way to set positives for deformed objects because the tracker will study more parameters from the background information, which affects the final tracking result. Pixel attention is proposed to solve this problem by keeping the training phase consistent with the tracking phase. In order to inherit the benefit of the pixel attention, we build a new Acc branch in the classification and regression head.

## 3. Method

This section will exhaustively introduce our PACR tracker in three aspects: the basic architecture, pixel-attention mechanism, and Acc branch. The whole network is shown in Figure 2, containing a Siamese network based on the CNN for feature extraction and a multi-head tracking network for object location. The prior network is similar to other Siamese networks, such as SiamCAR and SiamOA [37]. Meanwhile, a novel training method is adopted in the latter network.

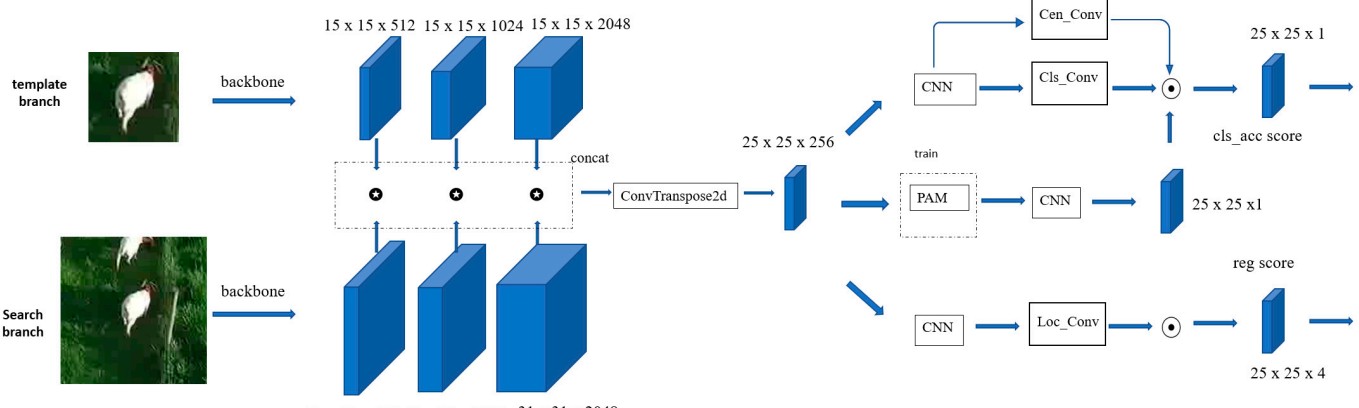

**Figure 2.** The entire network of PACR. The dimension of every model is shown near the model. We use concat and transposed conv to combine the feature extracted from the last three layers of the backbone. The PAM is the pixel-attention model.

*3.1. Basic Architecture*

The PACR divides the tracking task into two subtasks, similar to the classical Siamese-based tracker SiamCAR. One is extracting the feature from the tracking target. Another is obtaining the final target location with the help of the feature extracted from the first network.

As shown in Figure 2, the feature extracted network has two similar branches with the same parameter. The front is the search branch, and the bottom is the template branch. The PACR adopts the feature from layer 2, layer 3, and layer 4 of resnet50. The feature score $A^c_{w \times h \times c}$ will be obtained after the inverse convolution of the feature map, which is calculated by combining the feature extracted from the prior network. We adopt a novel design idea in the following classification prediction network. During the training and tracking phase, the PACR adopts two similar but not identical networks to ensure that our network has a high training speed and can also ensure that the performance of tracking is not significantly affected; the Acc branch was uniquely proposed to improve the regression results.

In the training phase, unlike the original design that directly uses the score $A^c_{w \times h \times c}$ related to the search frame and the template frame, we will input the related calculation score $A^p_{w \times h \times c}$ of the previous frame and the correlation score $A^c_{w \times h \times c}$ obtained from the current search frame and the template into the PAM (pixel-attention module), generate a related feature score that is more focused on tracking the target, input it into the Acc

branch, and input it into the other three branches to get four scores $\left(acc_{i,j}, cls_{i,j}, cen_{i,j}, loc_{i,j}\right)$, and then the total loss will be calculated with a joint classification–regression alternating refinement strategy combining several branches for optimization.

We get the coarse location generated by the above network:

$$pos_{coarse} = \arg\max_{i,j}\{(1 - \lambda_d)acc_{i,j} \times cls_{i,j} \times cen_{i,j} \times p_{i,j} + \lambda_d H_{i,j}\} \tag{1}$$

$H$ is the cosine window, $\lambda_d$ is a super-parameter, and the combination of these can suppress large changes in the bounding box and position. $p$ is a penalty updated with the search frame changed. After the coarse position is obtained, the accurate position $pos_{refine} = \left(x_{refine}, y_{refine}\right)$ will be calculated in a coarse-to-fine refine strategy [29]. $loc_{i,j}$ is a vector of $w \times h \times 4$ with each subvector representing the distance from the corresponding position to the four sides of the bounding box. So the height and width of the predicting box can be performed as follows:

$$
\begin{aligned}
w &= \frac{loc\left(x_{refine}, y_{refine}, 0\right) + loc\left(x_{refine}, y_{refine}, 2\right)}{2}, \\
h &= \frac{loc\left(x_{refine}, y_{refine}, 1\right) + loc\left(x_{refine}, y_{refine}, 3\right)}{2}
\end{aligned}
\tag{2}
$$

The final predicting box generated in the tracking phase is

$$B_{pred} = (x_{refine} - \frac{w}{2}, y_{refine} - \frac{h}{2}, w, h) \tag{3}$$

The whole tracking process is shown in Figure 3.

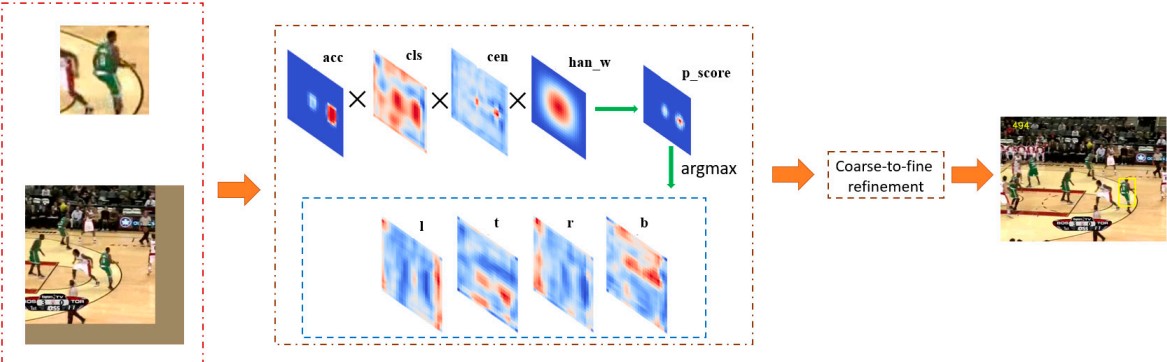

**Figure 3.** Detailed illustration of the tracking phase. The first rectangle is the picture of the search region and the template picture. The top four pictures indicate the acc score, cls score, cen score, and Hanning window, respectively. The bottom four pictures surrounded by the blue dot-line indicate the reg score.

### 3.2. Pixel Attention

The current method of determining the positive samples in the training phase of the anchor-free frame tracker is to directly learn all the features within a certain range of the ground truth bounding box, as shown in Figure 1. The sample from this method is unreasonable. Directly confirming the positive sample according to the distance from the center point will make the tracker learn more background information during tracking. Although this method generalizes the model of the tracking net, in the case of tracking targets with obvious changes in characteristics such as human movement, the reg score and cls score will be disturbed by the background due to sufficient learning of the background region, which will make the tracking result worse. More importantly, when the anchor-free tracker generates the bounding box in the tracking phase, it uses the maximum position of the correlation operation between the search frame and the template frame, and finally

determines the boundary according to the reg score of the maximum position. Taking the method mentioned above in training will lead to inconsistency between the tracking and training phases because the position of the maximum response point is considered a positive sample during tracking for the anchor-free tracker.

Above all, the tracker should determine the more accurate target area as positive samples such as the deep color area in Figure 1. We use a simple encoder named pixel attention, similar to a self-attention method in STMT. However, different from STMT, pixel attention uses the self-attention module to obtain the weight of each pixel so that the tracker can be trained. When we can pay more attention to the target, instead of fitting part of the background, the idea of pixel attention is closer to PSA [38], using self-attention to obtain the attention of each pixel. The PACR focuses on improving the precision of positive samples, helping enhance the consistency of the tracker in the training phase and the tracking phase. The pixel-attention module (PAM) is shown in Figure 4.

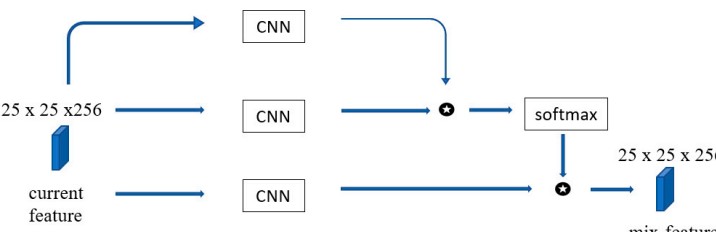

**Figure 4.** The framework of the pixel-attention module. The input is the correlation of the template feature and the search feature.

The main function of pixel attention is to make the tracker fit the position that is very likely to be the target point during the training phase and suppress the score of the non-target point area. The formula of the PAM is

$$w_{i,j} = \frac{\exp[(f_{i.}^q \otimes f_{.j}^k)/s]}{\sum_{\forall k} (f_{k.}^q \otimes f_{.j}^k)/s}, f_i = (f^v)_i^T \otimes w \tag{4}$$

The feature map $f^q, f^k, f^v$ is generated from three different Conv layers, and $w_{i,j}$ is the weights of the corresponding points, where $i$ is the index of each pixel on $f^q \in R^{THW \times C}$, and $j$ is the index of each pixel on $f^q \in R^{C \times HW}$. Here, $s$ is a scaling factor to prevent the exp function from overflowing numerically. Following [39], we set s to $\sqrt{C}$, where $C$ is the feature dimensionality of $f$.

### 3.3. ACC Branch

The PAM has two functions: one is to get a better positive sample into the trainer for training; another is to strengthen the feature and improve the ability to express the feature. To fully exploit the advantages of the PAM module in the tracking phase, we propose the Acc branch, which can optimize the scores generated by other branches through the joint classification–regression alternating refinement method from SiamOA [37]. In this branch, our tracker adopts the strategy of this paper [40]. The ideal result of the Acc branch is shown below:

$$Ac(i,j) = \Phi\left(\frac{P_{i,j}^3 + P_{i,j}^1 - 2 \times P_{i,j}^2}{(P_{i,j}^3 + P_{i,j}^1 - 2 \times P_{i,j}^2)_{avg}}\right) \tag{5}$$

$P^i$ represents the distance of the point of the same position from the bounding box in frame i, and for the moving trend is significantly greater than the movement trend of other

positive sample averages. It can be considered that this point is most likely the background. Otherwise, it is always considered as the tracking target. $\Phi(x)$ is performed as

$$\Phi(x) = \begin{cases} x & 0 \leq x \leq 1 \\ \frac{1}{x} & x \geq 1 \\ 0 & other. \end{cases} \tag{6}$$

The Acc branch learns the movement trend of the target, and by combining acc score and cen score, the final feature map can be closer to the real situation. The new version of the PAM is shown in Figure 5.

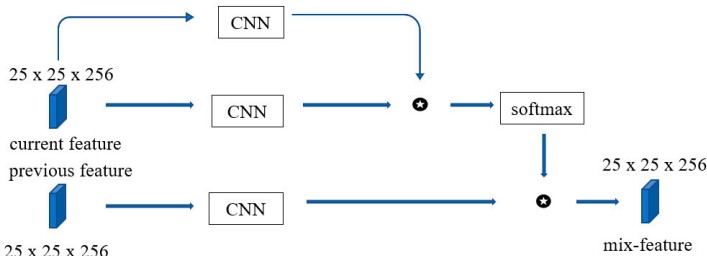

**Figure 5.** The upgraded pixel-attention module, compared with the old version. The input of this module changed to use both features of the previous frame and features of the current frame.

The input is changed from only the correlation score $f_c$ of the current search frame and the template frame to the correlation score $f_p$ of the previous frame and the template frame and the correlation score $f_c$ of the current frame and the template frame. The reason for our design is that the moving distance of the target is limited between two frames, so using $f_p \odot f_c$ to obtain $w$ will more accurately describe the tracked target. The loss of this branch is shown below:

$$\mathcal{L}_{acc} = \frac{-1}{\mathcal{N}} \sum Ac(i,j) \times \log A^{acc}_{w \times h \times 1}(i,j) + (1 - Ac(i,j)) \times \log(1 - A^{acc}_{w \times h \times 1}(i,j)) \tag{7}$$

By combining the score of Acc and the score of Cen, PACR can track the target occluded by other objects, as shown in Figure 6.

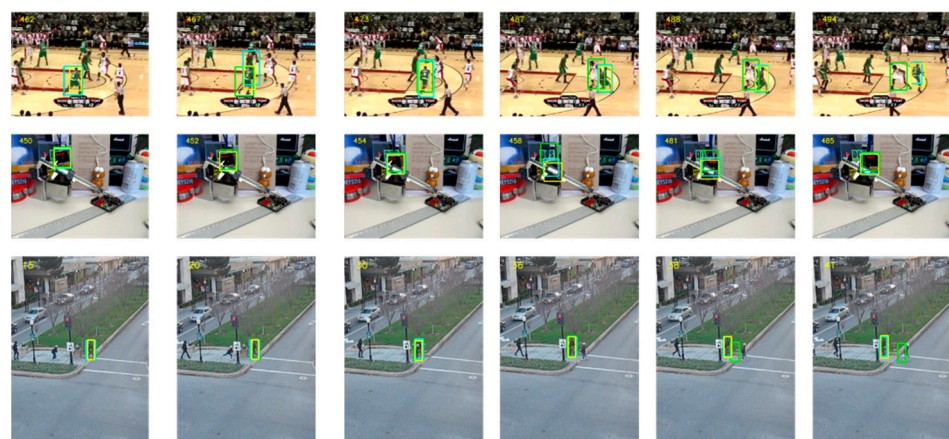

**Figure 6.** The result of tracking the occluded target compared with other trackers. The yellow line is our tracker, the green line is SiamCAR, and the light-blue line is the STMTracker.

## 4. Experiment and Results

### 4.1. Tracking Detail

Our network is implemented in Python with PyTorch and trained on 4 RTX3090 GPUs. The proposed PACR applies the ResNet50 as the feature extraction backbone with

the parameter pre-trained on ImageNet for the classification task. During the training phase, we adopt the PAM with the batch size set as 256 and epochs set as 50. For fair comparison [14,29], the parameter of the backbone network is frozen in the first 20 epochs. For the later 30 epochs, the last three blocks of ResNet50 are unfrozen for training. Specifically, our training datasets include COCO [41], imageNet VID [42], LaSOT [43], ImageNet DET [42], and YouTube-BB [44] for experiments on UAV123 [45], DTB70 [46], VOT2020 [47], and OTB100 [48]. The whole training phase takes about 56 h. For the large-scale experiments of LaSOT, we train our network in another 20 epochs with the training dataset LaSOT. We froze the parameters unfrozen in the first ten epochs and unfroze them in the later ten epochs. In the testing phase, following the idea we mentioned above, the PAM does not need to be taken part. The super-parameter which can be found in our code is just coarse-tuned. For evaluating the tracking results, we take advantage of the measurements from their official website, which may differ from each other.

### 4.2. Results on VOT2020

The VOT dataset is an authoritative dataset in single visual object tracking. The VOT2020 [47] dataset contains 60 public sequences with different challenging factors. VOT2020 has two different challenges: VOT-ST2020 and VOT-RT2020. The former is the VOT short-term tracking challenge under appearance variation, occlusion, and clutter. The latter is VOT short-term real-time challenge VOT-RT2020 under time constraints. The main metrics to evaluate the tracker are EAO (Expected Average Overlap), A (Accuracy), R (Robustness), and AUC (Average Unsupervised Accuracy). We compare our tracker with some typical and SOTA trackers from previous years. The results of our tracker are shown in Table 1

**Table 1.** The result of the VOT2020 dataset of PACR compared with other trackers. RT means real-time challenges. IVT (Incremental Visual Tracking), ATOM (Accurate Tracking by Overlap Maximization).

| | A | R | EAO | AUC | A(RT) | R(RT) | EAO(RT) |
|---|---|---|---|---|---|---|---|
| IVT [49] | 0.345 | 0.244 | 0.092 | 0.096 | 0.349 | 0.229 | 0.089 |
| KCF | 0.407 | 0.432 | 0.154 | 0.178 | 0.154 | 0.406 | 0.434 |
| SiamRPN++ | 0.443 | 0.672 | 0.244 | 0.229 | 0.443 | 0.673 | 0.244 |
| ATOM figure [50] | 0.462 | 0.734 | 0.271 | 0.378 | 0.440 | 0.687 | 0.237 |
| SiamFC | 0.418 | 0.502 | 0.179 | 0.229 | 0.422 | 0.479 | 0.172 |
| SiamCAR | 0.449 | 0.732 | 0.273 | 0.357 | 0.449 | 0.732 | 0.272 |
| PACR | 0.456 | 0.723 | 0.262 | 0.374 | 0.506 | 0.680 | 0.280 |

Compared with SiamCAR, our model baseline, we improve by 1.1% and 1.7%, respectively, in accuracy and AUC. Additionally, from Table 1, our tracker is better in accuracy, which indicates our PAM can help the tracker get more target information in the training phase, helping the tracker track better in the testing phase.

### 4.3. Results on OTB100

OTB100, proposed in 2015, is a classical benchmark in single visual object tracking containing 100 sequences with substantial variations collected and annotated from the commonly used tracking sequence. It has different attributes labeled with video sequences, such as deformation, motion blur, out-of-view, background clutter, etc. The metric of OTB100 is the precision plot and success plot. The former indicates the distance between the center of the bounding box and the center of the ground truth, and the latter means the IOU scores of the ground-truth boxes and the bounding box.

We compared our tracker with current mainstream methods, including anchor-free based trackers such as SiamCAR, SiamBan, Ocean, and TCTrack. The result has shown that our tracker is better than other anchor-free-based trackers. More information about the comparison is shown in Figure 7

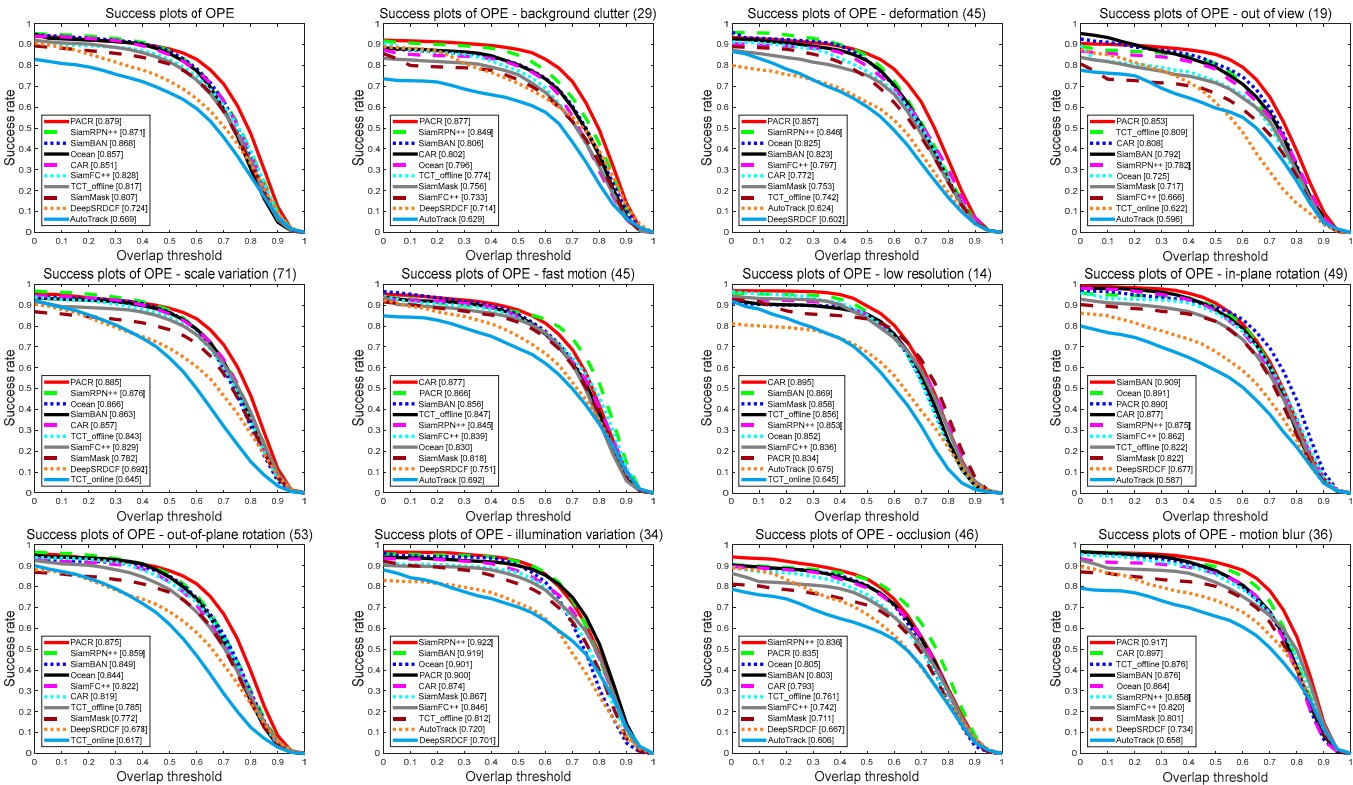

**Figure 7.** The result on different dimensions of OTB100 using OPE to evaluate.

## 4.4. Results on UAV123

UAV123 contains 123 sequences extracted from 91 videos and more than 110K frames. The sequences from this dataset are fully annotated. The same as OTB100, it also has many challenges for tracking, such as illumination variation, occlusions, and fast motion. The best difficult challenge in this dataset is the scale variation. We compare it with some state-of-the-art trackers in the UAV123 benchmark, and as with the OTB100 evaluation, we compare it with other anchor-free trackers.

The result of the UAV123 dataset evaluation is shown in Figure 8. Our tracker outperforms all other anchor-free-based trackers in precision, as the figure shows. Compared with CAR, our tracker improves the score by 1.8% in precision plots of OPE.

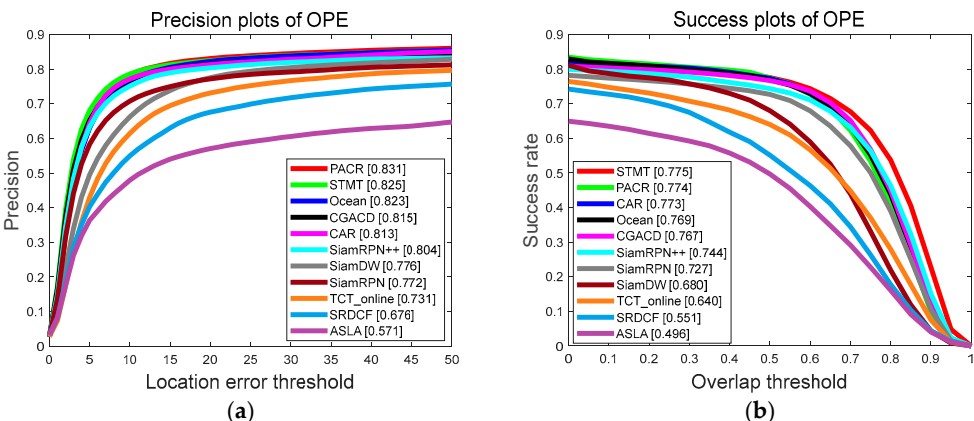

**Figure 8.** The evaluation results on the UAV123 dataset compared with some anchor-free trackers such as STMT, SiamCAR, Ocean, etc. (**a**) The precision points indicate the distance from the center point of the ground truth to the center point of the calculation by the trackers; (**b**) the success points indicate the overlap between the ground truth and the prediction.

### 4.5. Results on DTB70

DTB70, similar to UAV123, which is composed of 70 challenging UAV image sequences, primarily proposes the problem of severe UAV motion. The DTB70 dataset has tracking problems, such as various cluttered scenes and objects of different sizes. We compare our tracker with some SOTA trackers in this dataset, such as TCTrack and AutoTrack. The results come from their paper.

As shown in Figure 9, compared with TCTrack, the state-of-the-art UAV tracker in 2022, our tracker improves by 1.4% in success and decreases by 0.3% in precision. PACR has 55 FPS in this dataset which is much faster than TCTrack. Compared with CAR, our tracker obtains a 4.3 percentage point increase and one percentage point increase, respectively, in success and precision.

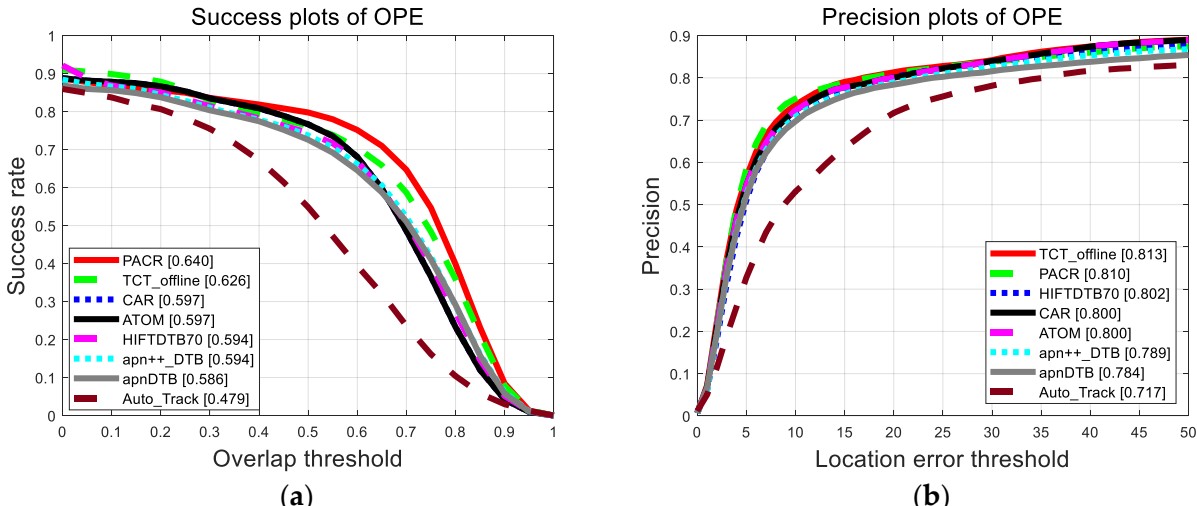

**Figure 9.** The result plot of DTB70 compared with TCTrack, SiamCAR, ATOM, etc. (**a**) The success points indicate the overlap between the ground truth with the prediction; (**b**) the precision points indicate the distance from the ground truth to the prediction.

### 4.6. Results on LaSOT

LaSOT is a large-scale single object tracking dataset with high-quality annotations. The dataset contains over 3.52 million manually annotated frames and 70 classes, each with 20 tracking sequences. Its training set consists of 1400 long videos, and its testing set consists of 280 long videos, with an average of 2500 frames per video. LaSOT can test the robustness of trackers properly, which have many difficulties tracking challenges such as occlusions, out-of-view, etc. The PACR is tested on the testing set. As shown in Table 2, compared with other trackers whose data are from their papers, PACR has shown better performance to these trackers. Our tracker also gets 65 FPS on average, which is much better than other trackers.

**Table 2.** The result of the evaluation on LaSOT.

|  | ECO | SiamRPN++ | AToM | TCTrack | SiamCar | PACR |
|---|---|---|---|---|---|---|
| Norm-precision | 0.338 | 0.569 | 0.576 | 0.484 | 0.610 | 0.674 |
| Auc | 0.324 | 0.496 | 0.515 | 0.435 | 0.516 | 0.581 |
| Precision | 0.301 | 0.491 | 0.505 | 0.414 | 0.524 | 0.587 |

### 4.7. Complexity Analysis

Compared with other trackers, such as SiamRPN++, our model has fewer parameters. The total FLOPS of SiamRPN++ is 59.56G, which is more than our model. The complexity of each part of our model is shown in the Table 3.

**Table 3.** The model complexity of each module which is calculated by PyTorch.

|          | Memory   | FLOPS    | MAdd     | MemR + W  |
|----------|----------|----------|----------|-----------|
| Backbone | 306.11 M | 44.66 G  | 89.27 G  | 772.59 M  |
| Neck     | 5.64 M   | 883.19 M | 1.76 G   | 25.06 M   |
| PAM      | 2.18 M   | 414.9 M  | 830.61 M | 50.5 M    |
| ACRhead  | 14.67 M  | 5.92 G   | 2.96 G   | 40.07 M   |

In order to reduce the model complexity, some small tricks [51] were used during the training and tracking phases. The first trick is that data enhancement performed on the images to ensure that the model's generalization ability is not affected by some adjustments to the resolution of the input images. Secondly, transfer learning is used to reduce the model training time with improved training efficiency and generalization ability of the model. Third, the total number of parameters has been reduced using Siamese structures at the backbone. Next, quantization is employed during training. Finally, a special structure named bottleneck is applied in the model's classification and regression part, reducing the number of parameters and the computational effort of the model.

*4.8. Discussion*

The results shown in the above datasets illustrate that our model gains excellent performance in certain scenarios, such as deformation challenges, scale variation challenges, background clutter challenges, etc., which accurately suggests that our module is effective. Our PAM ensures consistency between the tracking phase and training phase, leading to more accurate target finding in some turbulent scenarios.

In addition, the scores of the RT test on the VOT dataset indicate that our strategy of adopting different networks in the training and tracking phases is efficient, and the inference speed of classification and regression is significantly improved. Moreover, an unexpectedly exciting result arose: our results are much better on the UAV dataset than other anchor-free trackers.

However, the performance of our tracker in fast motion is not satisfactory compared to SiamCAR since we use a few frames close to each other as the target score for fitting. Similarly, it also impacts the robustness of tracking, which interprets that our R-score is poor compared to SiamCAR in the VOT baseline test. We will directly increase the range sizes between the frames to minimize errors in future experiments.

In conclusion, our experimental results can support that our PAM module and Acc branch are very effective in Single Object Tracking, and the PAM could significantly increase the performance of the anchor-free tracking network.

**5. Conclusions and Future Work**

A novel tracker named PACR is proposed to solve the inconsistency of object location rules between the tracking phase and training phase for Single Object Tracking. Our basic tracking framework is inspired by the Siamese-based networks. The PAM is proposed in our new tracker, and the output of the PAM will be sent to the Acc branch to adjust and fine-tune the result of the classification branch and regression branch, especially in some scenarios such as occlusion and deformation. Our tracker can achieve good results on OTB100, VOT2020, and LaSOT. Moreover, compared with other UAV trackers such as TCT and AutoTrack, our tracker achieves the best performance on DTB70 and UAV123.

In the future, we plan to apply a novel encoder–decoder architecture to our Acc branch, with the ConvNeXt being the backbone to extract the feature. During the ablation study, we tried to use only the PAM or the Acc branch in our tracker, and the results were worse than the comic SiamCAR. This is because the ideal result of the Acc branch requires the acceleration of the target.

**Author Contributions:** Conceptualization, D.L. and Q.W.; methodology, D.L. and H.C.; software, H.C. and Y.Z.; validation, H.C., Y.Z. and Y.X.; formal analysis, D.L. and H.C.; investigation, D.L.; resources, Y.X.; data curation, Q.W.; writing—original draft preparation, D.L. and H.C.; writing— review and editing, D.L. and Q.W.; visualization, H.C.; supervision, D.L.; project administration, Q.W. All authors have read and agreed to the published version of the manuscript.

**Funding:** This research was funded by the National Natural Science Foundation of China (NSFC) grant number 61801340.

**Institutional Review Board Statement:** Not applicable.

**Data Availability Statement:** The code could be available at https://github.com/wernety/PACR, accessed on 17 February 2023.

**Conflicts of Interest:** The authors declare no conflict of interest.

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
