# Peer review of "PACR: Pixel Attention in Classification and Regression for Visual Object Tracking"

_mathematics, doi:10.3390/math11061406_

Round 1
Reviewer 1 Report
1. The resolution of figures 2-5 and 8 is poor. Please update it with high-resolution figures.
2. In table 1, please also list the FPS values for other works to compare.
3. Please compare and discuss the computational complexity of your idea with other works in the literature.
4. Some advanced techniques are used to reduce the computational complexity and memory usage of neural networks, such as network pruning, weight quantization, etc. Have you used these techniques in your proposed neural network? Please briefly discuss it.
[1] Tang, Zhimin, et al. "Automatic sparse connectivity learning for neural networks." IEEE Transactions on Neural Networks and Learning Systems (2022).
Author Response
Dear Reviewer,
Thank you for taking the time to review my manuscript and for providing valuable feedback. Your comments and suggestions have been extremely helpful in improving the quality of my work.
I appreciate your thorough evaluation of the manuscript and your constructive criticism. Your expertise in the field has greatly contributed to improving the paper.
I have carefully considered all of your comments and made revisions. We have carefully considered all comments from the reviewers and revised our manuscript. First, We have made some grammatical revisions to the paper. Then we changed the article’s title and added some necessary data in the abstract. Next, We replaced some ambiguous words, such as this method appearing in the introduction section. And we have also extended some abbreviations when they first appeared. Then, we add 4.6 to explain the complexity and 4.7 to discuss our tracker. Finally, we double-checked our References and added some, such as compared tracker mentioned in section.4(Experiment and Results).
Point 1: The resolution of figures 2-5 and 8 is poor. Please update it with high-resolution figures
Response: We thank the reviewer for raising this recommendation. We have updated the image resolution to improve clarity and ensure the figures are more easily readable. We appreciate your help in ensuring the quality of our submission.
Point 2: In table 1, please also list the FPS values for comparing other works.
Response: We are so grateful for your kind recommendation. In the VOT2020 test, the official evaluation of real-time performance uses RT challenges[1] instead of FPS. The official evaluation method can reduce errors due to different hardware. We supplemented the RT challenges in Table 1. (paper 8, line 273 in clear_version.docx)
Point 3: Please compare and discuss the computational complexity of your idea with other works in the literature.
Response: We appreciate the reviewer for this kind recommendation. As far as we know, computational complexity is often used to measure the effectiveness of an algorithm in solving problems. However, when we consulted relevant literature about the tracker mentioned in our manuscript, we found that only some algorithms discussed and compared computational complexity in papers. So we can only calculate the complexity of our model in section 4.6(paper 10 line 335 - paper 10 line336 in clear_version.docx) and hope that more similar work will appear in the future.
Point 4: Some advanced techniques are used to reduce the computational complexity and memory usage of neural networks, such as network pruning, weight quantization, etc. Have you used these techniques in your proposed neural network? Please briefly discuss it.
Response: Thank you for your question and recommendation. To simplify the computational complexity and memory usage of neural networks, we takes some tricks such as Feature selection, model architecture design and bottleneck layer. We discuss them in section 4.6(paper 11 line 336 – paper 11 line 344 in clear_version.docx ). And we have cited the paper you provide.(paper 13, line 483 in clear_version.docx)
Once again, I would like to express my sincere gratitude for your efforts and for providing such a detailed and insightful review. I look forward to hearing from you soon.
- Matej, K., Ales, L., Jiri, M., Michael, F., Roman, P., The eighth visual object tracking vot2020 challenge results. In ECCV, 2021; pp 547-601.
Best regards,
Mr. Chai

Reviewer 2 Report
Dear Authors,
The main concerns are given as follows:
· There are grammatical and punctuation errors in the paper. The authors require a native speaker to proofread. The authors can use the professional version of the Grammarly system.
· The title of the paper should be extended clearly.
· The Abstract should be improved by adding the values of the best results.
· In the following sentence, what word does this method refer to in the introduction section?
*** This paper will interpret why we put forward this method and how we overcome the inconsistencies.
If you did not mention it in the previous paragraph, write the name of the method in the relevant sentence.
· The authors should review more studies by 2023.
· The authors should assign plots of the Accuracy and ROC for methods, and also should present values in a table.
· Abbreviations have not been placed in the text of the paper. For example, the ACRT is mentioned for the first time at the beginning of the Method section, but its extended word was not observed. Also, abbreviations of IVF, KCF, SiamPRN, ATOM, etc should be reformed correctly based on the mentioned structure.
· The title of Table 1 should be corrected in the following:
***Table 1. The result on the VOT2020 dataset of ACRT compared with other trackers
· The resolution of all Figures should be increased to 200 dpi.
· The authors should include the datasets links in the footnote.
· The discussion section should be written before the last section. The study’s strengths, weaknesses, and limitations should be written in a separate section after the discussion section.
· The conclusion section should be explained more clearly and shortly and also future work should be written.
Author Response
Dear Reviewer,
Thank you for your prompt review of my manuscript. I appreciate the time and effort you took to provide thoughtful and constructive feedback.
I am grateful for the comments you have made and carefully considered each of them. Your feedback has helped me improve my work’s quality and address areas that require further attention. We have carefully considered all comments from the reviewers and revised our manuscript. First, We have made some grammatical revisions to the paper. Then we changed the article’s title and added some necessary data in the abstract. Next, We replaced some ambiguous words, such as this method appearing in the introduction section. And we have also extended some abbreviations when they first appeared. Then, we add 4.6 to explain the complexity and 4.7 to discuss our tracker. Finally, we double-checked our References and added some, such as compared tracker mentioned in section.4(Experiment and Results).
Comment 1: There are grammatical and punctuation errors in the paper. The authors require a native speaker to proofread. The authors can use the professional version of the Grammarly system.
Response: We regret there were so many problems with the English of our old manuscript. We have used the professional version of the Grammarly system to check our grammatical and punctuation errors. And a native English speaker has carefully revised the paper to improve the grammar and readability of the manuscript.
Comment 2: The title of the paper should be extended clearly
Response: We are so grateful for your kind recommendation. The old title is ACRT: Attention in Classification and Regression; the word Attention may cause some ambiguity, so we extended “Attention” to a vital innovation point in our article, “pixel attention.” Then the new title is PACR: Pixel Attention in Classification and Regression for Visual Object Tracking;(paper 1, line 1 in clear_version.docx)
Comment 3: In the following sentence, what word does this method refer to in the introduction section?
*** This paper will interpret why we put forward this method and how we overcome the inconsistencies.
If you did not mention it in the previous paragraph, write the name of the method in the relevant sentence
Response: We appreciate the reviewer for this kind recommendation and regret this problem. This method means our tracking algorithm. To express clarity and accurate, We replaced it with our tracker’s name(paper 2, line 54 in clear_version.docx)
Comment 4: The authors should review more studies by 2023.
Response: We gratefully appreciate your valuable suggestion. Lately, we have been studying Transformer architecture in Visual Tracking. We have decided to apply a novel encoder-decoder to replace the acc branch, which could significantly increase our tracking results.
Comment 5: The authors should assign plots of the Accuracy and ROC for methods and also should present values in a table.
Response: We are grateful for your kind recommendation. As far as we know, The ROC curve can reflect the recognition ability of the classifier. In some classifiers based on deep learning, the ROC curve can better reflect the classifier’s performance. But in Single Visual Object Tracking, we always use precision figures and success figures to evaluate trackers’ performance [1,2] rather than the ROC curve. As shown below:
![]() |
![]() |
The left figure is the success figure compared with other trackers in OTB100. The horizontal axis indicates the percentage of The overlapping regions of ground truth and predicted box divided by the value of the total area of ground truth and predicted box. The vertical axis indicates the percentage of predicted boxes that satisfy a certain overlap area. For example, point (1,0) indicates that the number of completely overlapping prediction boxes and ground truth boxes is 0 frames in total frames. This curve shows that when the overlap rate is from 0 to 1, the prediction box generated by our tracker conforms to the change of a certain overlap rate. The score in the legend is the AUC in Object tracking. The AUC can reflect trackers’ performance on average. The right figure is the precision figure compared with other trackers in OTB100. The horizontal axis indicates the distance between the ground truth's center and the predicted box's center. The vertical axis indicates the percentage of predicted boxes that satisfy a certain distance. For example, point (0,0) indicates that the center of the prediction box and the center of the ground truth completely coincide has 0 frames in total frames. So in object tracking, these figures can more directly reveal the performance of trackers than the ROC curve.
Comment 6: Abbreviations have not been placed in the text of the paper. For example, the ACRT is mentioned for the first time at the beginning of the Method section, but its extended word was not observed. Also, abbreviations of IVF, KCF, SiamPRN, ATOM, etc, should be reformed correctly based on the mentioned structure.
Response: We appreciate the reviewer for this kind recommendation, and we are very sorry for our incorrect writing. The ACRT is our proposed tracker, and we have extended it. (paper 2, line 54)We have extended the other abbreviations is other object trackers, which are typical and have had very high performance in that years. We had extended other abbreviations when they first appeared. (paper 8, line 275 - line 276 in clear_version.docx)
Comment 7: The title of Table 1 should be corrected in the following:
***Table 1. The result on the VOT2020 dataset of ACRT compared with other trackers
Response: We are grateful for your kind recommendation and regret this problem. We have revised the title of Table 1 according to your suggestions(paper 8, line 273 in clear_version.docx) and checked out all other titles of figures and tables.
Comment 8: The resolution of all Figures should be increased to 200 dpi.
Response: We are very sorry for the negligence of picture quality. We have increased the resolution of all graphics as far as we can. In addition, we rechecked the quality of all the graphics to ensure they were visible when enlarged.
Comment 9: The authors should include the datasets links in the footnote.
Response: We are so grateful for your recommendation. We have put all datasets links in the footnote and checked that they are all valid.(paper 1 and paper 7 in clear_version.docx)
Comment 10: The discussion section should be written before the last section. The study’s strengths, weaknesses, and limitations should be written in a separate section after the discussion section.
Response: Thank you for your rigorous recommendation. We have added a new section(4.7 paper 11, line 345 in clear_version.docx). We briefly discuss the result of the experiment and also discuss the strengths, weaknesses, and limitations before the last section. But we didn’t write in a separate section after the discussion section.
Comment 11: The conclusion section should be explained more clearly and shortly and also future work should be written.
Response: Thank you for pointing out this problem in the manuscript. We have cut down some sentences in the Conclusion and added some details to make the Conclusion clearer and more shortly. (paper 13, line 366-line379 in clear_version.docx)
I hope that the revisions I have made have addressed your concerns satisfactorily. If there are any remaining issues that you would like me to address, please do not hesitate to let me know.
Again, thank you for your time and feedback. I look forward to hearing from you soon.
- Wu, Y.; Lim, J.; Yang, M. H., Object Tracking Benchmark. IEEE Trans Pattern Anal Mach Intell 2015, 37, (9), 1834-48.
- Mueller M, S. N., Ghanem B, A benchmark and simulator for uav tracking. In ECCV, 2016; pp 445-461.
Best regards,
Mr Chai
